# [Au]/[Ag]-catalysed expedient synthesis of branched heneicosafuranosyl arabinogalactan motif of *Mycobacterium tuberculosis* cell wall

Shivaji A. Thadke[1,*], Bijoyananda Mishra[1,*], Maidul Islam[1], Sandip Pasari[1], Sujit Manmode[1], Boddu Venkateswara Rao[1], Mahesh Neralkar[1], Ganesh P. Shinde[1], Gulab Walke[1] & Srinivas Hotha[1]

Emergence of multidrug-resistant and extreme-drug-resistant strains of *Mycobacterium tuberculosis* (MTb) can cause serious socioeconomic burdens. Arabinogalactan present on the cellular envelope of MTb is unique and is required for its survival; access to arabinogalactan is essential for understanding the biosynthetic machinery that assembles it. Isolation from Nature is a herculean task and, as a result, chemical synthesis is the most sought after technique. Here we report a convergent synthesis of branched heneicosafuranosyl arabinogalactan (HAG) of MTb. Key furanosylations are performed using [Au]/[Ag] catalysts. The synthesis of HAG is achieved by the repetitive use of three reactions namely 1,2-*trans* furanoside synthesis by propargyl 1,2-orthoester donors, unmasking of silyl ether, and conversion of *n*-pentenyl furanosides into 1,2-orthoesters. Synthesis of HAG is achieved in 47 steps (with an overall yield of 0.09%) of which 21 are installation of furanosidic linkages in a stereoselective manner.

[1] Department of Chemistry, Indian Institute of Science Education and Research, Dr Homi Bhabha Road, Pune, Maharashtra 411 008, India. * These authors contributed equally to this work. Correspondence and requests for materials should be addressed to S.H. (email: s.hotha@iiserpune.ac.in).

**M**ycobacterium tuberculosis (MTb) is the causative agent of Tuberculosis, the deadly disease that is plaguing mankind[1–6]. Robert Koch noticed that MTb has a thick and waxy cellular envelope, which was later shown to not only act as a large obstruction to the entry of antibiotics but also modulate the host immune system[3,4]. Some of the currently administered frontline drugs are demonstrated to inhibit the biosynthesis of cell wall[7,8]. The complete structure of the cell wall of MTb has been unravelled to observe that it has two major components termed as mycolylara-binogalactan and lipoarabinomannan wherein arabinose and galactose are in furanosyl and mannose in the pyranosyl form[9–14]. Ara*f*- and Gal*f*- are xenobiotic to humans and, therefore, understanding the biosynthesis of the cell wall components containing them is of particular significance for developing novel therapeutic agents[12]. Prospect of biological significance coupled with the challenge of large oligofuranosides synthesis has cajoled many synthetic carbohydrate chemists to develop strategies for assembling AG and lipoarabinomannan fragments[15–29].

Synthesis of oligosaccharides is an art of its own and each synthesis poses a unique challenge and demands deployment of multiple glycosyl donors[30–32]. To date, the largest fragment of the mycobacterial arabinogalactan was synthesized independently by groups headed by Lowary and Ito; but without Gal*f* residues (Fig. 1)[33,34].

In this Article, we show that the highly branched heneicosa-furanosyl arabinogalactan (HAG) can be synthesized by the repeated use of gold-catalysed activation of alkynyl 1,2-*O*-orthoester chemistry that was developed in our laboratory[35–40].

## Results

**Retrosynthesis.** Retrosynthetic disconnections of heneicosafur-anoside **1** recognized that the assembly of heneicosafuranoside requires four major constituents namely cassettes A–D (Fig. 2). Heneicosafuranoside **1** can be synthesized by the 1,2-*trans* diastereoselective furanosylation between propargyl 1,2-orthoe-ster of tetrasaccharide **2** and the tridecasaccharide-aglycon under gold-catalysed glycosidation conditions. Synthesis of trideca-saccharide **3** can be envisaged from 1,2-orthoester of a hexasaccharide cassette B (**4**) and the heptasaccharide-aglycon **5**. Heptasaccharide synthesis can be realized by the gold-catalysed furanosylation between a tetraarabinofuranosyl orthoester cassette C (**6**) and the trisaccharide cassette D (**7**). Synthesis of cassettes A–D is envisioned from building blocks **8a–8d**, **9**, **10**. Propargyl 1,2-orthoesters are envisioned from corresponding *n*-pentenyl glycosides wherein *n*-pentenyl moiety serves as an

excellent protecting group that can be transformed to anomeric bromide in transit to the orthoester preparation[38,41].

**Synthesis of monosaccharide-building blocks.** HAG synthesis started with the development of methods for the large-scale synthesis of building blocks **8a–8d** (Fig. 3). Easily accessible 1,2-orthoester **11** (ref. 36) was saponified under Zemplén deacetylation conditions to afford a diol, which was treated with one molar equivalent of TBDPS-Cl to get the alcohol **12**. A portion of the compound **12** was esterified using BzCl/py to obtain building block **8b** (see Supplementary Figs 1–3) and the remaining portion was converted into disilyl ether **8d** (see Supplementary Figs 7–9).

Further, gold-catalysed glycosidation conditions[36] were employed on glycosyl donor **8b** to obtain *n*-pentenyl furanoside **13** followed by the deprotection of TBDPS group using Py·HF to obtain the desired building block **8a** in very high yields. Gold-catalysed glycosidation between orthoester **8d** and 4-penten-1-ol resulted into the *n*-pentenyl furanoside **14**, which was subsequently transformed into acetate **15**. Deprotection of silyl ethers in presence of Py·HF followed by esterification afforded the compound **16**. *n*-Pentenyl furanoside **16** was converted into the 1,2-orthoester-building block **8c** employing a recently established protocol[38]. Compound **16** was treated with Br₂/CH₂Cl₂/4 Å mass spectrometry (MS) at 0 °C for 15 min to obtain the glycosyl bromide that was immediately treated with propargyl alcohol and 2,6-lutidine to afford the building block **8c** in 85% yield over two steps (see Supplementary Figs 4–6).

Galactofuranoside **17** (ref. 42) was uneventfully converted into orthoester **9** in two steps followed by the gold-catalysed furanosylation to afford compound **18** that was converted into building block **10** in four steps namely saponification of compound **18** resulting into a tetraol, locking of C-5 and C-6 hydroxyls as isopropylidene using 2-methoxypropene/ p-Toluenesulfonic acid (PTSA) in dichloromethane, esterification of C-2 and C-3 hydroxyl groups, and cleavage of isopropylidene using PTSA in MeOH with an overall yield of 70% (see Supplementary Figs 13–15; Fig. 4). Conversion of *n*-pentenyl furanosides into propargyl 1,2-orthoesters is considered to increase the reactivity of the donor and also conduct the reactions under catalytic conditions[38].

**Synthesis of cassettes A–D.** Synthesis of Heneicosafuranoside commenced with the preparation of cassettes A–D. Towards this affect, one molar equivalent of Gal*f* orthoester **9** was added dropwise to a solution of diol **10** in CH₂Cl₂ and allowed to react under standard gold-catalysed glycosidation conditions to obtain

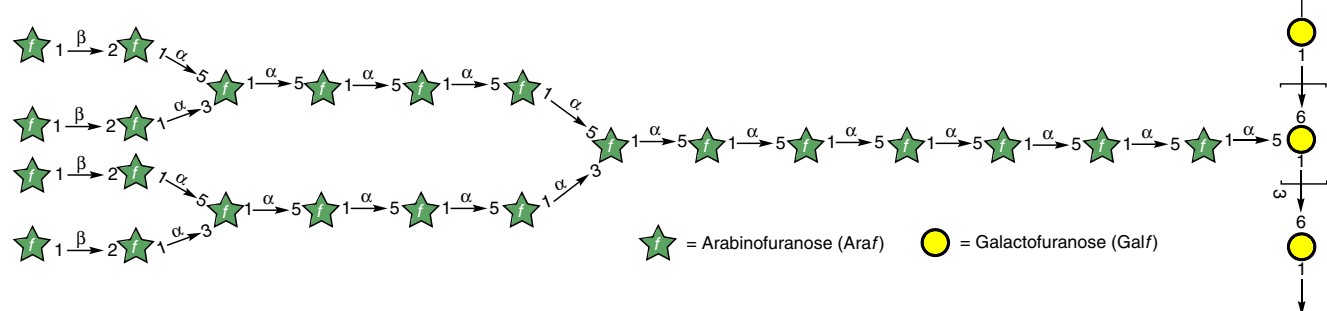

**Figure 1 | Arabinogalactan motif of *Mycobacterium tuberculosis* cell wall.** Arabinan is attached at the C-5 position of the galactan. Both arabinose and galactose are in the furanosyl form. Gal*f*(1→6)Gal*f* linkages and Ara*f*(1→3 or 5)Ara*f* linkages are all in the 1,2-*trans* manner, whereas Ara*f*(1→2)Ara*f* linkages are *1,2-cis* mannered.

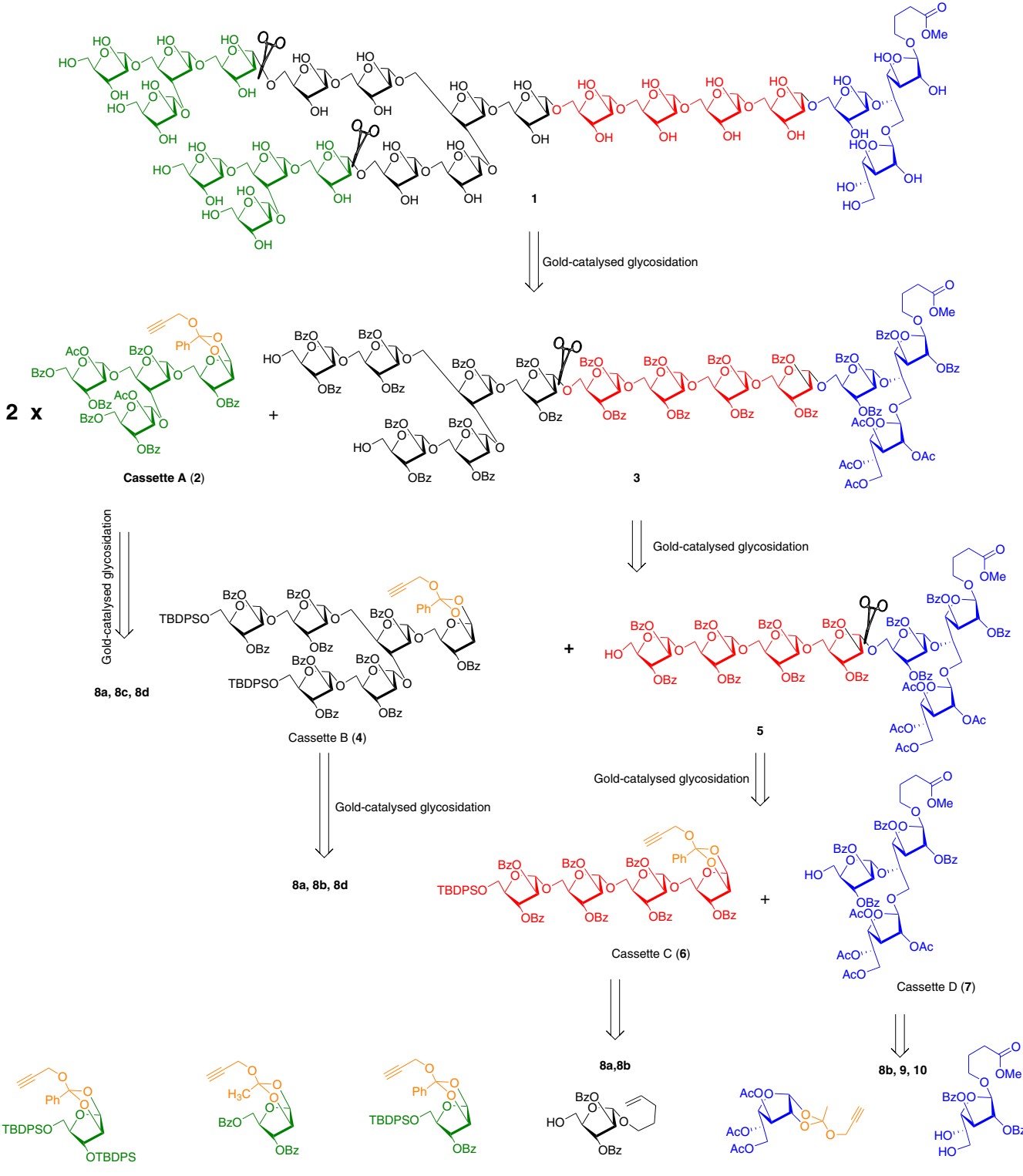

**Figure 2 | Retrosynthetic plan for the synthesis of heneicosafuranoside 1.** Retrosynthetic analysis shows the assembly of heneicosafuranosyl arabinogalactan (**1**) can be synthesized from cassettes A–D which in turn can be obtained from monosaccharides **8a**–**8d**, **9** and **10**.

the Gal*f*(1→6)β-Gal*f* disaccharide regioselectively in 70% yield. In continuation, the first arabinofuranosyl residue was attached at the *C*-5 hydroxyl group of disaccharide **19** under gold-catalysed conditions using the orthoester **8b** to afford the arabinogalactan **20** in 85% yield, which was converted to cassette D (**7**) by F-mediated cleavage of silyl ether (Fig. 5).

Synthesis of cassette C (**6**) commenced with the gold-catalysed glycosidation reaction between donor **8a** and aglycon **8b** in CH$_2$Cl$_2$ to obtain disaccharide **21** (ref. 38). Deprotection of silyl ether resulted in the aglycon **22** and the treatment of disaccharide **21** with Br$_2$ in CH$_2$Cl$_2$ followed by propargyl alcohol and 2,6-lutidine, TBAI afforded the glycosyl donor **23** in 85% yield. An

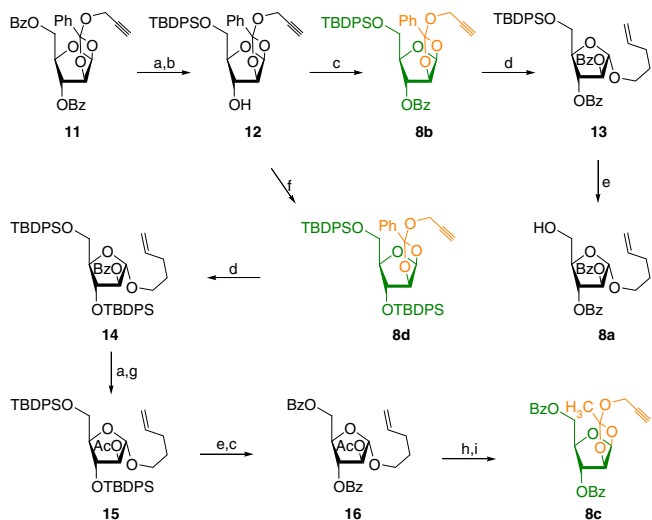

**Figure 3 | Synthesis of arabinofuranoside-building blocks.** Reagents and conditions: (a) NaOMe, MeOH, 25 °C, 6 h. (b) TBDPS-Cl (1 eq.), imidazole, *N,N'*-dimethylformamide (DMF), 0 → 25 °C, 12 h. (c) BzCl, Et$_3$N, CH$_2$Cl$_2$, 0 °C → 25 °C, 61%: three steps for **8b** from **11**. (d) Pent-4-enol, AuCl$_3$, 4 Å MS, CH$_2$Cl$_2$, 25 °C, 2 h. (e) Py · HF, THF:Py (5:1), 0 °C → 25 °C, 5 h, 66%: two steps for **8a** from **8b**. (f) TBDPS-Cl (1 eq.), Imidazole, DMF, 25 °C, 10 h, 74%: two steps for **8d** from **11**. (g) Ac$_2$O, pyridine, 0 °C → 25 °C. (h) Br$_2$, CH$_2$Cl$_2$, 4 Å MS, 0 °C, 15 min. (i) Propargyl alcohol, 2,6-lutidine, CH$_2$Cl$_2$, 4 Å MS, 0 °C → 25 °C, 10 h, overall 57% from **8d**.

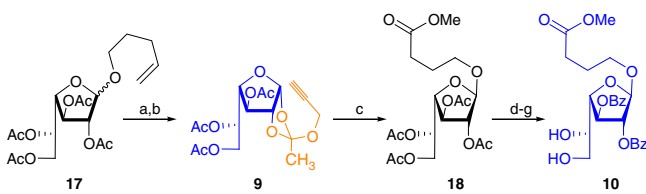

**Figure 4 | Synthesis of Galactofuranoside-building blocks.** Reagents and conditions: (a) Br$_2$, CH$_2$Cl$_2$, 4 Å MS, 0 °C, 15 min. (b) Propargyl alcohol, 2,6-lutidine CH$_2$Cl$_2$, 4 Å MS 0 °C → 25 °C, 10 h, 81% over two steps. (c) Methyl 4-hydroxybutanoate, AuCl$_3$, 4 Å MS, CH$_2$Cl$_2$, 25 °C, 2 h, 80%. (d) NaOMe, MeOH, 25 °C, 15 h. (e) 2-Methoxypropene, acetone, PTSA, 8 h. (f) BzCl, py., 0 → 25 °C. (g) PTSA, MeOH, 25 °C, 4 h, 70% over four steps.

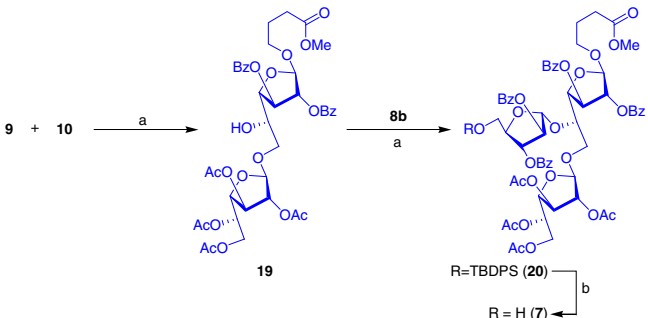

**Figure 5 | Synthesis of cassette D.** Reagents and conditions: (a) AuCl$_3$ (7 mol%), AgOTf (7 mol%), CH$_2$Cl$_2$, 4 Å MS powder, 25 °C, 2 h; 70% for **19** and 75% for **20**. (b) Py · HF, THF:Py (5:1), 0 °C → 25 °C, 4 h, 95%.

uneventful reaction between aglycon **22** and donor **23** employing AuCl$_3$/AgOTf conditions afforded the tetrasaccharide **24**, which was converted into cassette C (**6**) in two well-optimized steps (Fig. 6).

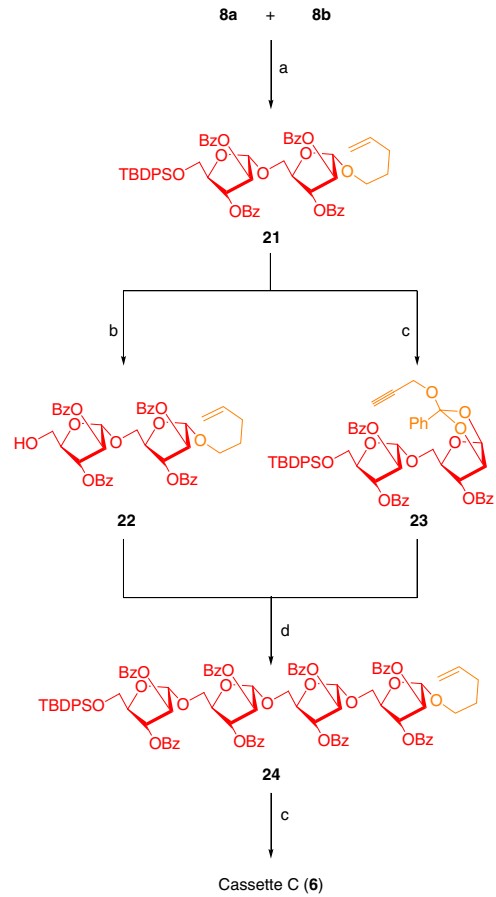

**Figure 6 | Synthesis of cassette C.** Reagents and conditions: (a) AuCl$_3$ (7 mol%), CH$_2$Cl$_2$, 4 Å MS powder, 25 °C, 2 h, 84%. (b) Py · HF, THF:Py (5:1), 0 °C → 25 °C, 4 h, 95%. (c) Br$_2$, CH$_2$Cl$_2$, 4 Å MS powder, 10 min, 0 °C followed by propargyl alcohol, 2,6-lutidine, tetrabutylammonium iodide, 0 °C → 25 °C, 12 h, 85% for **23** and 88% for **6**. (d) AuCl$_3$ (7 mol%), AgOTf (7 mol%), CH$_2$Cl$_2$, 4 Å MS powder, 25 °C, 12 h, 84%.

*En voyage* to the synthesis of heneicosafuranoside **1**, synthesis of cassettes A and B continued with the successful synthesis of disaccharide **25** from arabinofuranosyl donor **8d** and aglycon **8a**. *n*-Pentenyl disaccharide **25** was converted to diol **26** under Py · HF/THF/4 h/80% and reacted with donor **8b** (2.5 eq.) to obtain a tetrasaccharide **27**, which was converted into a diol that was immediately treated with 2.5 molar equivalents of donor **8b** to afford the hexasaccharide **28** commissioning gold-catalysed glycosidation conditions. Finally, conversion of *n*-pentenyl furanoside was converted into the cassette B (**4**) in two steps by reacting with Br$_2$/CH$_2$Cl$_2$/4 Å MS/0 °C/15 min followed by propargyl alcohol/2,6-lutidine/CH$_2$Cl$_2$/4 Å MS/0 → 25 °C/10 h in 86% yield over two steps. Similarly, diol **26** was treated with donor **8c** (2.5 eq.) to obtain a tetrasaccharide **29**, which was converted into cassette A (**2**) via glycosyl bromide intermediate (Fig. 7).

**Synthesis of HAG.** In continuation of this expedition, gold-catalysed glycosidation between donor cassette C (**6**) and acceptor cassette D (**7**) afforded the heptasaccharide (**30**) containing five Ara*f*- residues and two Gal*f*- residues in 1,2-*trans* disposition. The successful synthesis of heptasaccharide was confirmed by the $^{13}$C NMR spectral studies, wherein all seven anomeric carbons of compound **30** were noticed between 105.3 and 106.7 p.p.m. (see Supplementary Fig. 59).

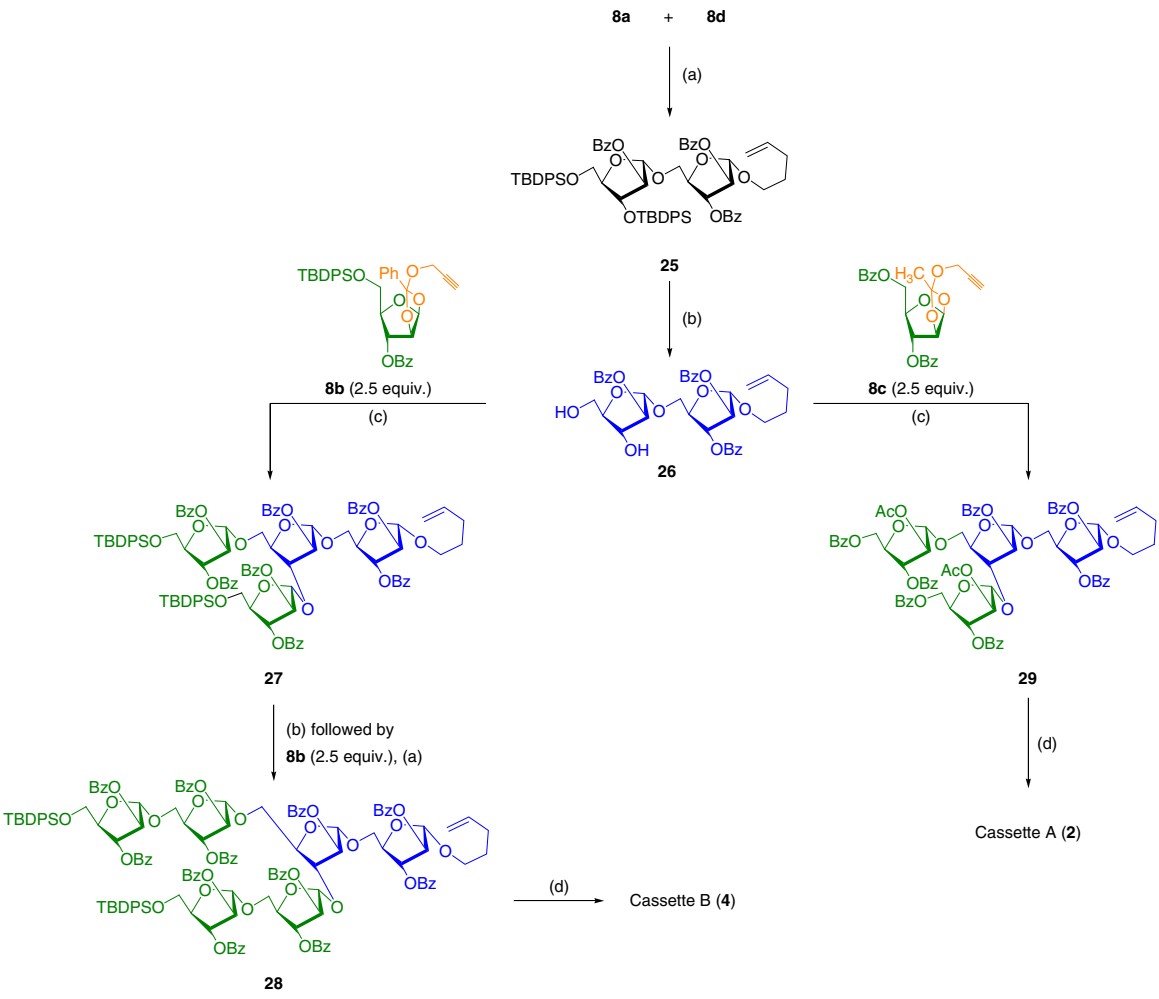

**Figure 7 | Synthesis of cassettes A and B.** Reagents and conditions: (a) AuCl₃ (7 mol%), CH₂Cl₂, 4 Å MS powder, 25 °C, 2 h, 81% for **25** and 50% for **28**. (b) Py · HF, THF:Py (5:1), 0 °C→25 °C, 4 h, 80% for **26** and 75% for **28**. (c) AuCl₃ (7 mol%), AgOTf (7 mol%), CH₂Cl₂, 4 Å MS powder, 25 °C, 2 h, 48% for **29** and 45% for **27**. (d) Br₂, CH₂Cl₂, 4 Å MS powder, 10 min, 0 °C followed by propargyl alcohol, 2,6-lutidine, tetrabutylammonium iodide, 0 °C→25 °C, 12 h, 88% for **2** and 86% for **4**.

The *C*-5 hydroxyl group at the non-reducing end of heptasaccharide **30** was unmasked by the addition of Py · HF in THF to obtain alcohol **5**, which was coupled with *n*-pentenyl donor **28** to obtain the required tridesaccharide **31** in 24% yield after 4 days. However, the reaction between heptasaccharide **5** and cassette B (**4**) under the standard gold-catalysed conditions afforded tridecaoligosaccharide **31** in 65% yield (Fig. 8).

Unmasking of the silyl ether using Py · HF/THF afforded the glycosyl acceptor **3** that can be glycosylated with a tetrasaccharide-donor cassette A (**2**). The gold-catalysed glycosidation between acceptor **3** and 2.5 equivalents of the donor **2** was performed at 25 °C for 24 h to afford the HAG (**32**) in fully protected form. In the ¹H NMR spectrum, all singlets between δ 4.97–5.75 p.p.m. indicated the presence of all 1,2-*trans* linkages at the anomeric position and in the ¹³C NMR spectrum, resonances due to 21-anomeric carbons appeared between δ 105.2–106.7 p.p.m. (see Supplementary Fig. 71). Further, matrix-assisted laser desorption/ionization–time of flight–mass spectrometry (MALDI–TOF–MS) also supported the formation of HAG (**32**) of MTb cell surface (Fig. 9). Zemplèn deacylation using 0.5 M NaOMe in MeOH afforded the fully deprotected HAG with methyl butanoate linker at the reducing end. In the ¹H and ¹³C NMR spectra, resonances due to the benzoate moiety completely disappeared. The 150 MHz ¹³C NMR spectrum

showed signals at δ 176.9, 52.2, 30.6 (−CH₂), 24.2 (−CH₂) p.p.m. confirmed the presence of the methyl butanoate linker at the reducing end. Resonances in the anomeric region did not resolve fully for complete assignment even at this field, although two resonances at δ 108.60 and 108.63 p.p.m. confirmed the presence of two β-Gal*f*- residues on the basis of previous assignments (see Supplementary Figs 72,73). In addition, the high-resolution MALDI–TOF mass spectrum of compound **1** showed a molecular ion of the sodium adduct at *m/z* = 2,974.2810 (see Supplementary Fig. 74), matching satisfactorily with that of calculated exact mass of compound **1**.

In summary, the synthesis of HAG containing nineteen 1,2-*trans*-Ara*f*s and two Gal*f*s was successfully achieved by utilizing Au/Ag-catalysed furanosylations in 0.09% overall yield. The key features associated with this effort are: (1) the stereoselective installation of 1,2-*trans* Ara*f*- and Gal*f*- residues using propargyl 1,2-orthoesters as glycosyl donors, (2) all glycosylation reactions were catalytic, high yielding and thus easy to purify, (3) the convergent fragment coupling between linear/branched oligosaccharides using propargyl 1,2-*O*-orthoester donor chemistry. Essentially, the expedient synthesis of HAG built on only three repetitive reactions namely glycosidation in the presence of catalytic amount of AuCl₃ and AgOTf; cleavage of *O*-silyl ether using Py · HF; and transformation of *n*-pentenyl

**Figure 8 | Synthesis of tridecaoligofuranoside.** Reagents and conditions: (a) AuCl₃ (7 mol%), AgOTf (7 mol%), CH₂Cl₂, 4 Å MS powder, 25 °C, 24 h, 70% for **30** and 65% for **31**. (b) Py · HF, THF:Py (5:1), 0 °C → 25 °C, 4 h, 85% (c) NIS (3 eq.), TfOH (0.3 eq.), 4 days, 0 °C → 25 °C, 24%.

glycosides into 1,2-orthoesters. The whole synthesis is modular and thus amenable for the synthesis of various saccharide mutants for eliciting the biological response.

## Methods

**General methods.** Unless otherwise noted, materials were obtained from commercial suppliers and were used without further purification. Unless otherwise reported, all reactions were performed under Argon atmosphere. Removal of solvent *in vacuo* refers to distillation using a rotary evaporator attached to an efficient vacuum pump. Products obtained as solids or syrups were dried under a high vacuum. Gold and silver salts were purchased from Sigma-Aldrich India Limited. Amberlite was purchased from Sigma-Aldrich and Bio-gel P-4 gel was purchased from Bio-Rad Laboratories, USA. Analytical thin-layer chromatography was performed on pre-coated silica plates (F₂₅₄, 0.25 mm thickness) from Merck; compounds were visualized by ultraviolet light or by staining with anisaldehyde spray. Optical rotations were measured on a JASCO 2000 P digital polarimeter. Infrared spectra were recorded on a Bruker Fourier transform infrared spectrometer. NMR spectra were recorded either on a Bruker Avance 400 or a 500 or

600 MHz with CDCl₃ or D₂O as the solvent and tetramethylsilane as the internal standard. High-resolution mass spectroscopy was performed using ABI MALDI–TOF or Waters Synapt G2 ESI mass analyser. Low-resolution mass spectroscopy was performed on Waters ultra performance liquid chromatography (UPLC)-MS, and thin-layer chromatography was checked on SWADESI-TLC MS interface. A small quantity (5–10%) of propargyl glycosides was observed in most of the gold-catalysed glycosidations with propargyl orthoesters as reported earlier[35]. For NMR analysis and high-resolution mass spectrometry of the compounds in this article, see Supplementary Figs 1–74.

**Gold(III)-catalysed 1,2-*trans* glycosidation.** To a CH₂Cl₂ solution (5 ml) containing glycosyl donor (0.1–1 mmol) and aglycon (0.1–1 mmol)) with 4 Å molecular sieves powder (0.1–1.0 g) was added a catalytic amount of AuCl₃ (7 mol%; AgOTf (7 mol%) as an additive wherever mentioned) and stirred at 25 °C (ref. 37). After 2 h (for oligosaccharides up to 24 h), the reaction mixture was neutralized by the addition of Et₃N and filtered through a pad of celite and concentrated *in vacuo*. The resulting residue was purified by silica gel column chromatography using ethyl acetate-petroleum ether to obtain 1,2-*trans* glycosides as a fluffy solids.

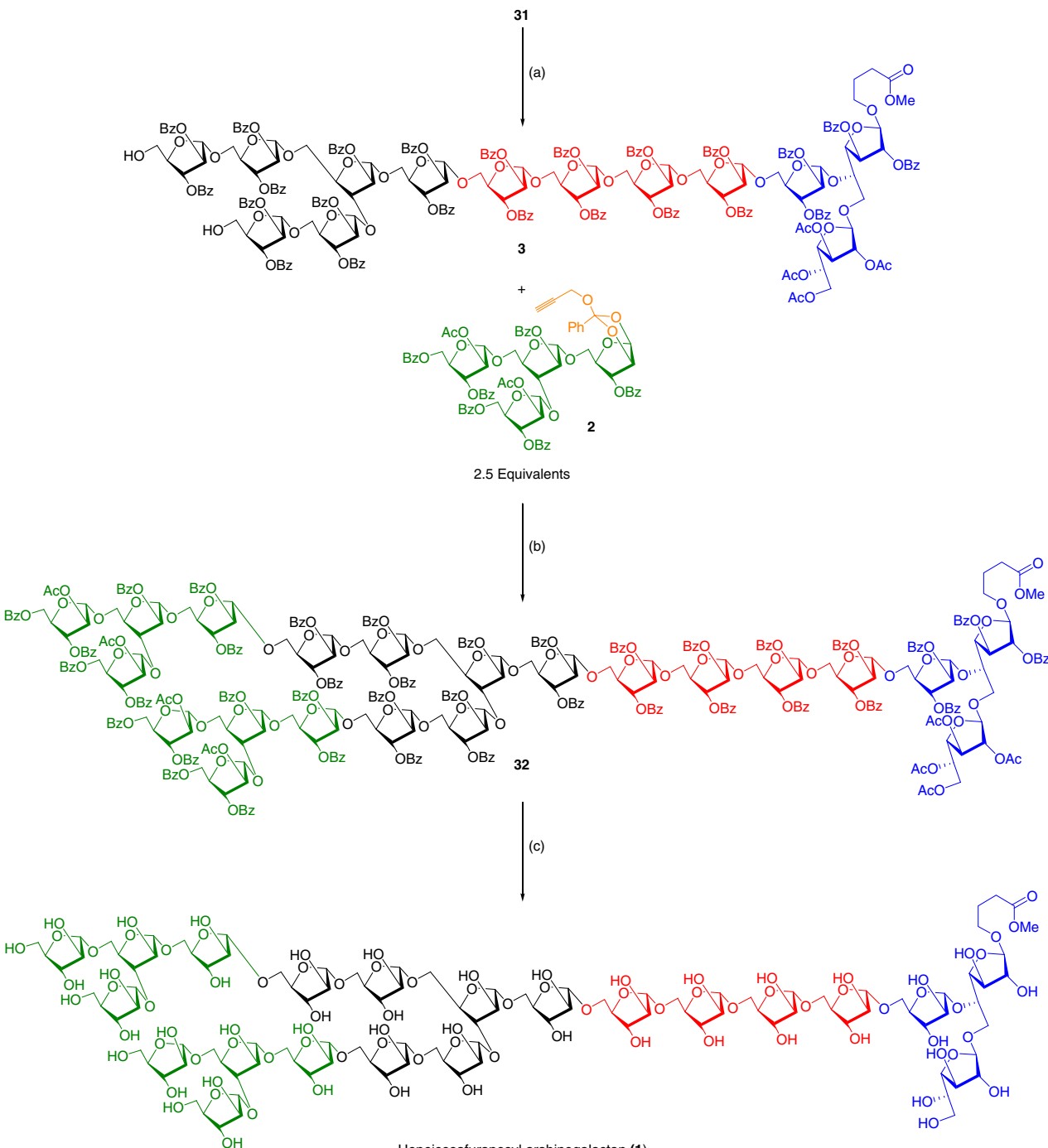

**Figure 9 | Synthesis of heneicosafuranosyl arabinogalactan (1).** Reagents and conditions: (a) Py · HF, THF:Py (5:1), 0 °C→25 °C, 4 h, 85%. (b) AuCl₃ (7 mol%), AgOTf (7 mol%), CH₂Cl₂, 4 Å MS powder, 25 °C, 24 h, 71%. (c) 0.5M NaOMe, MeOH, CH₂Cl₂, 4 h, 25 °C, 74%.

**Glycosidation by *n*-pentenyl glycosides.** To a CH₂Cl₂ solution (5 ml) containing glycosyl donor (0.1–1 mmol) and aglycon (0.1–1 mmol)) with 4 Å molecular sieves powder (0.1–1.0 g) was added 3 molar eq. of *N*-iodosuccinimide (NIS) at 0 °C in ice bath and stirred for 10 min[21]. After 10 min, catalytic amount of TfOH (0.3 eq.) was added to the reaction mixture at 0 °C and stirred at 25 °C. After complete disappearance of the donor (adjudged by TLC), the reaction mixture was neutralized by the addition of Et₃N and filtered through a pad of celite. The filtrate was washed with sat. aqueous solutions of sodium bicarbonate and sodium thiosulphate. Combined organic layers were dried over sodium sulphate and concentrated *in vacuo*. The resulting residue was purified by flash silica gel column chromatography using ethyl acetate-petroleum ether to obtain 1,2-*trans* glycosides.

**Pent-4-enyl glycosides to propargyl orthoesters.** Pent-4-enyl furanoside (1–50 mmol) was dissolved in anhydrous CH₂Cl₂ (10–500 ml) and cooled

to 0 °C (ref. 38). Br₂ (1.1 molar eq.) in CH₂Cl₂ was added dropwise to the reaction mixture with constant stirring at 0 °C. In addition, the reaction mixture was stirred for 10 min at 0 °C and concentrated under reduced pressure to give furanosyl bromide as white foam, which was immediately used in the next step without further purification.

The crude furanosyl bromide was redissolved (10–500 ml) in anhydrous CH₂Cl₂, propargyl alcohol (1.5–2 molar eq.) and 2,6-lutidine (2–3 molar eq.). Catalytic amount of tetra *n*-butyl ammonium iodide was added to the reaction and stirred for 4 h to overnight at room temperature. The reaction mixture was diluted with CH₂Cl₂ (100–500 ml) and water (100–500 ml), and the aqueous layer was extracted with CH₂Cl₂ (2*x*), the organic extract was washed with saturated oxalic acid solution and saturated sodium bicarbonate solution. The organic phase was collected, dried over sodium sulphate and concentrated *in vacuo*. Crude residue of the orthoester was purified by silica gel column chromatography (EtOAc:petroleum ether) to obtain propargyl 1,2-*O*-orthoester as a white foam/solid.

**Cleavage of O-silyl ethers.** To a solution of O-TBDPS protected saccharide (0.1–10 mmol) in THF:py (10:2–100:20 ml) was added Py · HF (2 molar eq. per O-TBDPS) and the reaction mixture was stirred at 25 °C for 4–12 h (ref. 38). The reaction was arrested by adding saturated aqueous solution of $NaHCO_3$ and extracted with EtOAc. The EtOAc layer was dried over anhydrous sodium sulphate and concentrated *in vacuo*. The crude residue was purified by flash column chromatography (EtOAc:petroleum ether) using silica gel.

**Deprotection of benzoates.** A 0.5 M NaOMe in MeOH (2 ml) was added to a solution of compound **32** (110 mg, 15.4 μmol) in 1:1 MeOH-$CH_2Cl_2$ (4 ml) and stirred at 25 °C. After 4 h, the reaction mixture was quenched by the addition of Amberlite-IR120 ($H^+$) resin, filtered and the filtrate was concentrated *in vacuo* to obtain a residue that was washed sequentially with chloroform and ethyl acetate to remove majority of the methyl benzoate. The remaining residue was purified by column chromatography using Bio-gel-P4 gel (90–180 μm, exclusion limit 4,000 Da). The compound was collected using Millipore water, concentrated *in vacuo* and further lyophilized for 24 h to obtain the HAG **1** (34 mg, 74%) as a white solid.

**Data availability.** The authors declare that some of the data supporting the findings of this study are available in its Supplementary Information files. All data are available from the authors upon reasonable request.

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

ARTICLE

## Acknowledgements

S.A.T., B.M., M.I., B.V.R., M.N. and G.P.S. acknowledge the fellowship from CSIR-UGC-NET. S.H. and S.M. thank the DST-New Delhi for the Swarnajayanthi fellowship. S.H. and S.P. thank financial assistance from CEFIPRA-New Delhi. G.W. thanks IISER Pune for the fellowship. We thank DST-FIST funds for high-field NMR facility at IISER Pune.

## Author contributions

S.H. planned and supervised the research; S.A.T. performed most of the experiments together with B.M.; M.I., S.P., S.M., B.V.R., M.N., G.P.S. and G.W. performed additional experiments; S.A.T., B.M., M.I., S.P., S.M., B.V.R., M.N., G.P.S. and G.W. isolated and characterized the products; S.P. purified the final deprotected oligosaccharide. S.H., S.A.T. and B.M. co-wrote the manuscript.

## Additional information

**Competing financial interests:** The authors declare no competing financial interests.

