## [Peer Review File · Nature Communications]

Reviewers' comments:

Reviewer #1 (Remarks to the Author):

The paper by Hotha and co-workers represents a heroic effort showing the preparation of a protected 21-mer arabinogalactan from the cell envelope of *Mycobacterium tuberculosis*. In the past numerous groups have tried to synthesize fragments of this highly important oligoglycan. The groups of Lowary and Ito are noteworthy, but they did not reach this stage of such a huge branched structure with the galactofuranose residues attached. Although automated synthesis is able to reach longer sequences, but all these sequences were uniform and consisted of relatively simple glycosidic linkages; such complicated structures as presented here were not achieved by a solid phase approach to date. Thus, I am deeply convinced that solution-phase synthesis has still its advantages as this paper nicely demonstrates. The Au/Ag-catalyzed glycosylation methodology developed in the Hotha group was exploited and showed its utility even for difficult glycosylations and the assembly of such big structures. The retrosynthetic analysis elegantly dissects the target in smaller parts which are accessible in a straightforward manner. Even the couplings of the big subunits were achieved with astounding yields. All compounds were carefully characterized by modern analytical methods. The paper was well assembled and presents the results in a clear and precise way. The group needs to be congratulated to this amazing effort. This methodology will pave the way for deprotection, biological studies and a series of other arabinogalactans related to *Mycobacterium tuberculosis*. Thus, I most enthusiastically support the publication of this submission in *Nature Communications* after some very minor revisions noted below:

- a) It would be interesting to read about an overall yield.
- b) The word "protected" should be inserted in the Abstract and conclusion.
- c) Line 51: typo: heinecosa....
- d) The point between HF and Py should be in the middle of the line.
- e) Sometimes blanks are missing, e.g. mol%Au in several schemes and between number and h or number and days.
- f) Line 239: deprotection of O-silyl ethers
- g) I think ref. 31 is not suitable at this point since it deals with thiooligosaccharides. An Angewandte review of R.R. Schmidt (with Zhu) 2008 or 2009 might be more useful.

Overall, very nice work which should be published as soon as possible.

Reviewer #2 (Remarks to the Author):

The authors synthesized very successfully a heneicosamer of an arabinogalactan motif of *Mycobacterium tuberculosis*. This motif contains 15 alpha-(1-5)-linkages and 3 alpha-(1-3)-linkages between arabinofuranosyl residues and 3 closely related linkage types between arabinofuranosyl and galactofuranosyl residues. Because of this structural simplicity only a few readily available building blocks are required that are connected by the gold catalyzed activation of alkynyl 1,2-orthoesters, a procedure developed in the Hotha group. Therefore, the statements in lines 17-25 and 171-181 are not really correct: Any other glycosidation procedure would not require "multiple glycosyl donors", anomeric selectivity would be also based on neighboring group participation as in the present paper, due to mainly linkages to primary hydroxy groups highyielding reactions could be expected, etc.

To recommend publication of this paper in Nature Communications complete deacylation of the final product 32 with about 40 acyl groups has to be performed. In addition, byproduct formation in the glycosidation reactions has to be discussed.

Point-wise Response to Reviewers' Observations

(NCOMMS-16-14139)

Reviewer 1:

Thank you very much for observing our manuscript to be suitable for publishing in Nature Communications.

Q1) It would be interesting to read about an overall yield.

A1) We calculated to find that the overall yield is 0.09% and mentioned the same in the abstract and in the manuscript.

Q2) The word "protected" should be inserted in the Abstract and conclusion.

A2) We have not inserted because, we have successfully deprotected the oligosaccharide and hence, the word protected was not necessary.

Q3) Line 51: typo: heinecosa....

A3) We have corrected in the revised manuscript.

Q4) The point between HF and Py should be in the middle of the line.

A4) We have corrected in the revised manuscript.

Q5) Sometimes blanks are missing, e.g. mol%Au in several schemes and between number and h or number and days.

A5) We have corrected in the revised manuscript.

Q6) Line 239: deprotection of O-silyl ethers

A6) We have corrected it as : cleavage of O-silyl ethers

Q7) I think ref. 31 is not suitable at this point since it deals with thiooligosaccharides. An Angewandte review of R.R. Schmidt (with Zhu) 2008 or 2009 might be more useful.

A7) Thank you very much for the suggestion. We have replaced the reference accordingly. Also, we found that the title in one of the references was wrong; so, we corrected that as well.

Reviewer 2:

Thank you very much for observing our manuscript to be suitable for publishing in Nature Communications.

Q1) The statements in lines 17-25 and 171-181 are not really correct: Any other glycosidation procedure would not require "multiple glycosyl donors", anomeric selectivity would be also based on neighboring group participation as in the present paper, due to mainly linkages to primary hydroxy groups highyielding reactions could be expected, etc.

A1) We believe that deployment of multiple glycosyl donors will be required owing to the presence of branching points though all are 1,2-trans linkages only. The main advantage being that the strategy is amenable for the rapid synthesis by solid phase methods as it significantly reduces the number of

optimizations on polymeric beads since the chemistry will be same for all glycosidations. The utility of single donor chemistry to synthesize furanosides having 1,2-*trans* linkages of both Araf- and GalF- in C-2,C-3 and/or C-5 positions as well as all the branching points would certainly be advantageous. We have revised the manuscript accordingly. We thank honourable reviewer for this suggestion.

Q2) Complete deacylation of the final product 32 with about 40 acyl groups has to be performed.

A2) Thank you very much for the suggestion. We have now deacylated the final product **32** that demanded repetition of all 46 steps. This endeavour showed the robustness of the method that we adopted for the synthesis and we learned a lot about the purification aspects of highly water soluble oligosaccharides. This experience would not be possible without your comments. The same methodology will be useful for us in purifying other oligosaccharides which are currently at different stages of synthesis. We sincerely appreciate your suggestion.

Q3) In addition, byproduct formation in the glycosidation reactions has to be discussed.

A3) We mentioned this in the revised manuscript.

REVIEWERS' COMMENTS:

Reviewer #2 (Remarks to the Author):

The authors essentially followed the reviewer requests in the amended manuscript. Therefore, publication of the paper in Nature Communications is recommended after the following correction:

For the final compound (1) the calculated mass data in the S.I. seem to be wrong.

Point-wise Response to Reviewers' Observations

(NCOMMS-16-14139A)

Reviewer 2:

**Q1) The authors essentially followed the reviewer requests in the amended manuscript. Therefore, publication of the paper in Nature Communications is recommended after the following correction:
For the final compound (1) the calculated mass data in the S.I. seem to be wrong.**

A1) Thank you very much for observing this. We have corrected in the revised manuscript.

Thank you very much for observing our manuscript to be suitable for publishing in Nature Communications.